# A Literature Review on the Use of Aortic Allografts in Modern Cardiac Surgery for the Treatment of Infective Endocarditis: Is There Clear Evidence or Is It Merely a Perception?

**DOI:** 10.3390/life13101980

**Published:** 2023-09-28

**Authors:** Francesco Nappi, Thibaut Schoell, Cristiano Spadaccio, Christophe Acar, Francisco Diniz Affonso da Costa

**Affiliations:** 1Department of Cardiac Surgery, Centre Cardiologique du Nord, 93200 Saint-Denis, France; tiboschoell@hotmail.com; 2Cardiothoracic Surgery, Lancashire Cardiac Center, Blackpool Victoria Hospital, Blackpool FY3 8NP, UK; cristianospadaccio@gmail.com; 3Department of Cardiothoracic Surgery, Hôpital Pitié-Salpêtrière, Boulevard de Hôpital 47-83, 75013 Paris, France; c.acar@aphp.fr; 4Department of Cardiovascular Surgery, Instituto de Neurologia e Cardiologia de Curitiba—INC Cardio, Curitiba 81210-310, Parana, Brazil; fcosta13@mac.com

**Keywords:** infective endocarditis, aortic valve, allograft, reinfection

## Abstract

Infective valve endocarditis is caused by different pathogens and 60% of those involve the aortic valve with valve failure. Although *S. aureus* is recognized as the most frequently isolated causative bacterium associated with IE in high-income countries, Gram-positive cocci nevertheless play a crucial role in promoting infection in relation to their adhesive matrix molecules. The presence of pili on the surface of Gram-positive bacteria such as in different strains of *Enterococcus faecalis* and *Streptococcus* spp., grants these causative pathogens a great offensive capacity due to the formation of biofilms and resistance to antibiotics. The indications and timing of surgery in endocarditis are debated as well as the choice of the ideal valve substitute to replace the diseased valve(s) when repair is not possible. We reviewed the literature and elaborated a systematic approach to endocarditis management based on clinical, microbiological, and anatomopathological variables known to affect postoperative outcomes with the aim to stratify the patients and orient decision making. From this review emerges significant findings on the risk of infection in the allograft used in patients with endocarditis and no endocarditis etiology suggesting that the use of allografts has proved safety and effectiveness in patients with both pathologies.

## 1. Introduction

Infective endocarditis (IE) is a pathological condition where an infectious process affects one or more valve leaflets, potentially spreading to adjacent structures of the heart and resulting in escalating clinical severity. Several pathogenic agents can trigger IE, with bacteria being the most commonly diagnosed type. Several studies have found that the annual incidence of IE ranges from three to nine cases per 100,000 people, with 60% of patients experiencing aortic valve (AV) damage and concomitant failure as a result of infective endocarditis. In high-income countries, IE is a leading cause of valve regurgitation [1,2,3,4,5,6,7].

Some studies have indicated that aortic valve involvement in IE is more common than mitral valve involvement. Hussain et al. [8] found consistent rates in 775 patients, with 51% of them developing IE with AV involvement; of those patients, 59% developed prosthetic valve endocarditis (PVE) and 68% developed invasive endocarditis. In contrast, MV involvement occurred in 30.7% of patients, with PVE and invasive endocarditis rates of 29% and 35%, respectively. Disparities in numerical percentage arise from a study conducted by Misfield and colleagues [9]. The study shows that AV is more involved than MV, with 47% for AV and 31% for MV.

Curlier et al. [10] found that the incidence of infective endocarditis (IE) varies from 43.8% to 35.4%. Worry about IE sustained by an active process affects both the natural and prosthetic valves, with different incidences-around 50% for native valve endocarditis (NVE) and 10% for prosthetic valve endocarditis (PVE)—as reported in the primary surgical series of two referral centers of experience. The pathoanatomical lesion in IE may affect one or more heart valves, varying in the degree of extension to different areas of the hear [11,12,13,14].

More aggressive infections caused by active processes can result in extensive and destructive damage to all components of the aortic valve, including the leaflet, annulus, aortic root, and left ventricular outflow tract [15,16,17,18,19,20]. Similarly, in patients with extended IE to the aorto-mitral intervalvular fibrosa or multiple valve involvement, which is reported in about 40% of cases [11,12,13,14], there is a significant increase in preoperative and postoperative mortality [1,2,3,4,5,21].

A higher in-hospital mortality rate (15% vs. 8.4%), as well as higher 6-month mortality (23% vs. 15%) and 1-year mortality (28% vs. 18%) rates were observed in patients with supported IE from S. aureus compared to those with non-S. Aureus infection. The former group was identified by the presence of extensive lesions, highlighting the need for prompt identification of the pathogen to achieve a timely diagnosis and prevent worsening of the primary infectious fields [5,18,21]. Typically, the pathogen is detected in approximately 90% of cases after three rounds of blood cultures, enabling the customization of antimicrobial treatment [22]. Specific serological tests for Bartonella, Coxiella burnetii, and Brucella are necessary in patients who belong to specific risk categories and have negative blood cultures. Additionally, certain strains of S. aureus and Streptococcus viridans may grow markedly aggressive and remain undetectable with common serological tests during infections, necessitating the use of more accurate identification methods such as mpB genotyping systems and matrix-assisted laser desorption/ionization time-of-flight systems [23].

*Enterococcus faecalis* is a common bacterium found in the human gastrointestinal and biliary tracts. However, it has gained significant attention due to its pathogenic capability causing infections in surgical sites, urinary tract, and bloodstream, hence becoming one of the main causes of such infections [24,25,26]. Similarly, group D streptococci, such as *Streptococcus gallolyticus* and *Streptococcus bovis*, which have been reclassified as *Enterococcus* spp., are known to promote infective endocarditis alongside the onset of gastrointestinal and urogenital tract disorders. This suggests that the portal venous system may serve as a gateway [27,28,29].

The degree of valve regurgitation has a significant impact on the progression of IE, which has an unpredictable natural course and is linked to the emergence of heart failure symptoms, pulmonary hypertension, and atrial fibrillation [9]. Endocarditis involving the aortic valve presents with lesions of varying degrees, characterized by destruction of the annulus and root, as well as embolic manifestations that can be localized in the coronary and cerebrovascular districts. This can potentially cause complications such as stroke and left ventricular dysfunction. In cases of endocarditis involving the mitral valve, there may be dysfunction of the posteromedial papillary muscle with varying degrees of injury to the cords, which can lead to left ventricular rupture [30,31,32,33]. Various population-based reports have shown that patients who underwent surgery had an in-hospital mortality rate of 24%, while those receiving medical therapy had a rate of 26.4%. After 1 year, the mortality rate for the latter group climbed to 29% [1,2,3,4]. The factors impacting mortality include age, shock, prosthetic valve endocarditis (PVE), left ventricular ejection fraction below 40%, and recurrent endocarditis [1,2,3,4]. Significant predictors of worse long-term follow up are age, shock, PVE, left ventricular ejection fraction below 40%, and recurring endocarditis affecting mortality [11]. Although the approach to treating and managing IE varies and is based on individual experiences, the established rise of multidisciplinary teams for collaborative decision making may alleviate this trend.

Allografts were widely used as the primary biological replacement for aortic valves [34], known for their favorable hemodynamics and low risk of thromboembolic complications [35]. However, early reports described varied techniques for collecting, decontaminating, preserving, and managing viable allograft cells, as well as for immunology [35,36]. The subcoronary implantation technique was the most common initial method used for allografts universally. Today, standardized cryopreserved allografts can be obtained commercially, and surgeons employ the more replicable insertion technique that encompasses root replacement. Although aortic allografts gained notoriety as an ideal substitute for low infection risk due to landmark studies published in 1992, which revealed an absence of an early peaking phase of recurrent endocarditis when compared with the elevated risk of other prostheses during this phase [37,38], there is little knowledge regarding freedom from early and long-term allograft endocarditis after allograft root replacements.

Our team has gained expertise in utilizing allograft aortic root implants for the treatment of any aortic valve and root diseases, encompassing non-invasive and invasive endocarditis, since 1992. While recent studies limit the use of allografts for non-endocarditis pathologies, we support the use of allogenic tissue in cases of native or prosthetic valve infections where the disease process is marked by infectious invasion beyond the aortic leaflets (invasive endocarditis). This encompasses situations with destroyed aortic root that hinder simple valve replacement [13,39].

Through literature review, we have developed a personalized strategy for managing endocarditis, considering clinical, microbiological, and pathological factors that impact postoperative results. The aim is to stratify patients and encourage shared decision making. Finally, we believe that utilizing a coordinated multidisciplinary approach in the diagnostic evaluation of IE is crucial for patients who require early surgical intervention, particularly in situations of high-risk embolization and/or clinical decline despite appropriate antimicrobial therapy. 

## 2. Methods

We searched MEDLINE, Embase, and the Cochrane Library using the search terms “endocarditis” or “infective endocarditis” together with “allograft”, “homograft”, “allograft endocarditis”, “pathogenesis”, “manifestations”, “treatment”, and “surgery”. We primarily chose publications from the past 10 years but did not exclude widely referenced and highly considered major publications. We also retrieved the reference lists of articles identified by this search strategy and selected articles that we judged relevant. Recommended review articles are included in tables and references alongside the text, providing readers with more details and background references (Table 1).

## 3. Results

Aortic valve infective endocarditis is a critical condition that demands a multidisciplinary effort to ensure consistent outcomes with its diagnosis and management [5]. While surgical replacement of the infected valvular tissue remains the primary treatment [19], the surgical community has been debating the best option for an aortic substitute, whether it should be a conventional xenograft or mechanical prosthesis, or allogenic or autologous tissues for decades. Several variables, such as the patient’s age, risk factors, comorbidities, and contraindications to anticoagulation, as well as anatomopathological factors such as the extent of the infection and pathogen virulence, the timing of intervention, and specific features of the aortic substitute used such as durability and risk of infection recurrence, require prompt assessment during the decision-making process for surgical treatment of IE [40,41,42,43,44,45,46,47,48,49,50,51].

### 3.1. Extent of Bacterium-Dependent Infective Endocarditis and Treatment Options for Native or Prosthetic Valvular Endocarditis

In over 90% of cases, IE of the aortic valve is caused by *Staphylococci*, *Streptococci*, and *Enterococci* spp. The proportion of involved pathogens may vary and is influenced by several determinants, including age, coexisting comorbidities, primary focus of infection, and type of valve involved. These factors differentiate native valve endocarditis (NVE) from more severe IE, which involves prosthetic valves (PVE). Nonetheless, oral beta-hemolytic *streptococcus group A* is detected significantly less frequently in high-income countries. On the other hand, *Streptococcus viridans* confirms its role in favoring a high percentage of IE as well as presenting a danger in active endocarditis. *S. aureus* is commonly isolated as the main cause of IE in high-income countries, accounting for up to 30% of all infection cases.

On the other hand, *enterococci* spp., identified as normal intestinal commensals, have emerged as opportunistic pathogens and should be considered as causative agents [1,2,3,6]. *Enterococci* have become one of the leading causes of nosocomial bloodstream, surgical site, and urinary tract infections, with *E. faecalis* being the most clinically relevant among *Enterococcus* spp. It accounts for approximately 5–8% of hospital-associated bacteremia and approximately 5–20% of all cases of endocarditis [52]. Infective endocarditis caused by *E. Faecalis* is a condition where the heart’s valves or inner lining become infected, causing valve damage and potential fatality in the absence of effective antibiotic therapy. The emergence of resistance to multiple antibiotics has emphasized the significance of acquiring new understandings into enterococcal endocarditis, a critical clinical challenge which highlights the necessity for alternative antibiotic strategies [53,54,55,56]. In contrast to *group A streptococci* or *S. aureus*, which are highly pathogenic microorganisms with a specificity that correlates to the production of numerous hemolysins and toxins aimed at effectively neutralizing innate immune responses [57,58], the pathogenesis of *E. faecalis* is less dependent on virulence factors [59].

Emerging evidence shows that the first step in the infectious process of *E. faecalis* is primarily induced by the attachment and colonization of host tissue surfaces [59,60]. In support of this hypothesis, we have identified mainly Gram-positive pathogens suggesting the vital role played by proteins of the family of adhesive matrix molecules (MSCRAMM), which may serve as potential antigenic candidates for the development of new, promising immunotherapies [61]. Scientists have recently confirmed the presence of pili on the surface of Gram-positive bacteria, including various strains of *Streptococcus* spp., *Actinomyces* spp., and *Corynebacterium* spp. [62,63,64]. The existence of adhesins on the tip of the pilus, as well as the major and minor subunits of the pilus, has been proven through the use of antibodies as detection reagents [65]. Sillanpää and colleagues [66] proposed producing antibodies against pili proteins detected more often in the sera of *E. faecalis*-infected patients than in sera from uninfected control patients [66].

The primary issue concerns negative blood cultures, which can meet 10% of the Duke criteria standards. This clinical condition presents two plausible explanations: (1) The pathogen may be concealed by antibiotic therapy commenced before or at the same time as the planned blood culture series. (2) Fastidious bacteria play a role in the development of infective endocarditis, as evidenced by cultures positive for *Bartonella* spp., *Brucella* spp., *Coxiella burnetii*, and a group of bacteria identified as HACEK (*Haemophilus species*, *Actinobacillus*, *actinomycetemcomitans*, *Cardiobacterium hominis*, *Eikenella corrodens*, and *Kingella kingae*). The use of Polymerase Chain Reaction (PCR) assays on valve and blood samples, along with other microbiological diagnostic methods, can help identify pathogens that are difficult to detect. This results in an unmasking rate of 60% in cases of Infective Endocarditis (IE). It should be noted that certain strains of *Staphylococcus aureus* and *Streptococcus viridans*, known for their aggressiveness, are able to elude commonly used tests. The mBP genotyping and matrix-assisted laser desorption/ionization time-of-flight system can detect these pathogens. However, the discovery of microorganisms can sometimes take too long, leading to worsened lesion evolution. Negative cultures must also be taken into account as they can delay the start of antimicrobial treatment and negatively impact the evolution of infectious fields, particularly in cases of active IE. This is illustrated in the graphical abstract. Given the disease progression, a delay of 24–48 h in antibiotic delivery may exacerbate the patient’s clinical condition, warranting consideration of emergency surgery [1,2,3,29] 

Internists often require extended time to achieve a diagnosis due to the challenge of identifying the causative pathogen. Concerns are related to the presence of aggressive and non-detectable microorganisms which may necessitate 24 to 48 h of non-targeted antibiotic therapy, leading to the extension of lesions with multiple valve involvement and destruction of large portions of the heart, resulting in adverse outcomes. For instance, this situation can be identified in infectious endocarditis promoted by intracellular microorganisms such as *C. burnetii*, *Bartonella* spp., or *Tropheryma whipplei*, wherein the exposure and immune response status of the host become critical [1,2,3,7,67,68,69] (refer to the graphical summary).

### 3.2. Clinical Use of Allograft

In recent years, growing evidence has indicated that surgical aortic valve replacement (AVR) for adult infective endocarditis is frequently ineffective. Even after receiving an aortic root replacement using an allograft, the possibility of allograft infection persists. While allografts have a positive history in endocarditis, this physiological mechanism remains poorly comprehended, compared to the high risk of other prostheses during this stage, thereby diminishing its lifespan.

The option of allogeneic tissue has been debated by many surgeons due to the risk of encountering another disease by treating one, specifically due to early structural valve degeneration (SVD). Therefore, the decision-making process when selecting an allograft over traditional stented xenograft or mechanical prosthesis is pivotal. The surgery using the allograft option is technically challenging, resulting in in-hospital mortalities ranging from 2 to 5.5% during elective surgery, making it an acceptable option compared to other prostheses as reported in the literature [46,47,48,49]. However, emergency surgery mortality rates increase for SVD or relapse of infection [39,41,42,70,71].

Surgeons may be hesitant to use allografts due to the lack of clinical benefits demonstrated in randomized clinical trials. However, a single randomized trial has shown that the Ross procedure is safer and more effective than aortic allograft implantation, resulting in better survival, improved quality of life, and increased durability. It should be noted that this trial only included a small number of patients with endocarditis. Several long-term studies investigating the use of allografts in comparable patient populations have shown improvements in survival rates even up to 20 years later [39,41,42,43,44,68,69,72,73]. However, it should be noted that direct conclusions drawn from these observational studies should be approached with caution. This is due to the fact that the selection of patients who received an allograft, for both non-endocarditis indications and infectious endocarditis, may explain some of the differences in outcomes observed. This has renewed interest in the possible role of allogeneic tissue in IE. Several studies have shown better survival rates and fewer valve infections in patients who received allografts compared to those who had prosthetic valve replacement in complex aortic valve surgery for IE. Allogeneic tissue recipients had a 2% incidence [39,41,43,68,69] versus 25.4% in conventional valve recipients during the early postoperative period [41]. It should be noted that multiple observational reports have shown the occurrence of clinical and echocardiographic evidence of recurrence for infectious events within the first year [11,12,41,44].

Active endocarditis is a well-established risk factor for both early [11,12,39,41,44] and late mortality [11,12,39,41,44,68,69,73]. Differences in long-term outcomes might be attributed, in part, to the need for lifelong anticoagulants after implanting a mechanical prosthesis. This increases the risk of bleeding, stroke, and thromboembolism, which are severe long-term complications [74]. Several reports, submitted by institutions with over 10 years’ experience and based on strong patient cohorts, recommend the use of allogeneic tissue in patients with active infective endocarditis of the aortic valve [41,42,68,69,74]. Based on these experiences, it is more probable to utilize allogeneic tissue in patients with active endocarditis and those with infectious fields caused by particularly aggressive bacteria. The studies referenced [11,12,39,41,44,68,69,73,75,76,77] support this conclusion.

Yankha and colleagues [42] showed outstanding clinical and durability outcomes in aortic root allograft reconstruction for endocarditis with perianular abscess. The survival rate at 1 and 10 years was 97% compared with 91%, with an extremely low rate of reinfection of allogenic tissue. Over 20 years of experience from Musci and colleagues [41] has shown that a significant proportion of recipients (*n* = 1163) who received an aortic allograft for complex endocarditis with perianular abscess (*n* = 221) had a lower rate of reinfection (5.4%) for both NVE and PVE. Additionally, freedom from reoperation was greater than 92% at 10 years for both groups. Although there is a significant metric disparity in early and late mortality when comparing NVE and PVE, high rates of death are reported in both groups (16% vs. 25% and 37% vs. 45%). This suggests the complex nature of the procedure in patients with clinical conditions, which may invite criticisms.

Fukujima et al. [69] analyzed the largest cohort of patients in the Brisbane group who underwent allograft valve replacement [74]. They reported a low incidence of reinfection during the 30-day and mean follow-ups of 5 years, ranging from 16 days to 16 years.

Arakbani et al. [68] similarly reported a reoperation rate of 2.2% at 27 years due to allograft infection. Moreover, antibiotic treatment effectively treated allograft infections in patients. Further evidence of discrepancies in outcome measures for reoperation caused by infection or SVD is common in the literature. 

The Cleveland Clinic team [43] reported outstanding clinical outcomes for endocartion rate in both groups but acknowledged that the structural degeneration of allografts was an inevitable issue, which led them to opt for a bioprosthesis. For instance, Sabik and coauthors [76] investigated a sample of 103 patients and found that those who required an allograft for root reconstruction in PVE had an operative mortality rate of 3.9%. Moreover, 97% of these patients did not experience reinfection more than two years after undergoing allograft implantation.

Nonetheless, Bekkers and colleagues [78] found a 3.84% reoperation rate for reinfection after 14 years, although freedom from homograft root re-operation due to structural valve degeneration reached 59.3% after the same period. Kowert et al. [70] have noted a 24.8% allograft recovery rate due to infection (15%) or SVD (85%) after 15.5 years. Although technically demanding, re-operations can be carried out with satisfactory outcomes. However, patients with post-allograft infections had increased mortality rates [39,43,68,69,70,71].

We present our findings on the utilization of aortic allografts in 51% of patients with infective endocarditis, and in 24% of cases involving a primary infective process of the mitral valve or with secondary extension to intervalvular fibrosa for invasive endocarditis. Two-thirds of the patients received an aortomitral monobloc, while the remaining third received a separate block with partial mitral allograft. The technique of implantation has been successful in cases where the heart tissue has been weakened by infection, with good results evident during a 20-year follow-up. This evidence suggests that allografts are more likely to be implanted in active endocarditis with extensive lesions [79].

### 3.3. The Antibacterial Properties of Allograft

Following recent evidence, the adoption of a polyester graft at the site of infective endocarditis has gained consensus among cardiac surgeons as it leads to acceptable results. These results have been directly correlated to the heightened efficacy of antibiotic treatment. Thus, the use of antimicrobial drugs alone to treat certain cases of PVE has reduced the necessity for allograft transplantation [80]. However, it is important to carefully consider the benefits of medical treatment before ruling out surgery [1,2,3,7]. Therefore, the potential benefit of removing prosthetic material in an infected area may be less convincing in present times, but it is dependent on the heart team’s expressed direction for the joint multidisciplinary decision-making process. Steffen and associates recommend employing allogeneic tissue in infective endocarditis instances affecting significant portions of the heart in both NVE and PVE. The severity and extent of infectious lesions are critical factors in determining whether to use the cryopreserved allograft, as this valve replacement maintains antibacterial properties even after being stored for up to 5 years [81]. The study found that applying a combination of antibiotics, including gentamicin, piperacillin, vancomycin, metronidazole, amphotericin B, flucloxacillin, meropenem, tobramycin, and colistin, to allogeneic tissue during the preservation process significantly affects its resistance to prolonged infection from both Gram-positive and Gram-negative bacteria. Certain strains of staphylococcus (*S. epidermidis* and *S. aureus*) showed significantly reduced infectivity in ascending aortic allograft tissue, with less bacterial contamination found in the ascending aorta than in the aortic valve leaflets. As a result, the allograft tissue’s infection resistance was considerably enhanced. Flucloxacillin was administered to ensure protection against Pseudomonas aeruginosa, whilst meropenem and colistin were utilized for *E. coli* [81].

Although antibiotics used after thawing of allografts lead to a significant reduction in infection recurrence [82], this effect is not verified for conventional prostheses or Dacron graft treatment. Nevertheless, the risk of vascular graft infection seems to decrease with antibiotic pretreatment of the graft [82]. Additionally, the antibiotic/fibrin compound has been shown to have a positive impact on preventing early infection relapse by facilitating the delayed release of antibiotics [83]. Furthermore, recent observations have indicated that higher concentrations of β-lactam antibiotics may provide an extra antimicrobial defense in the event of an active infection recurrence [83]. It has been demonstrated through numerous key series that administering antibiotics to allografts yields a positive outcome, with success rates ranging from 21% to 25% [12,68,69,73].

The study recently published by Witten et al. [43] is timely and examines the risk of allograft infection in patients who have received allografts for indications other than endocarditis and in those who have had infective endocarditis. The researchers analyzed the instantaneous risk of allograft infection using a time-varying method. They employed a parametric multiphase temporal decomposition, nonproportional hazards and machine learning analysis on patients who underwent aortic valve replacement (AVR) with aortic allografts from 1987 to 2017. A total of 2042 patients underwent allograft aortic valve procedures at a quaternary medical center with 53% of cases for non-endocarditic indications and 47% for endocarditic indications, 68% of which included prosthetic valve endocarditis (PVE). The mean age for patients undergoing allograft implants was 52 ± 14 years in the cohort without endocarditis and 57 ± 15 years in the cohort with endocarditis. The leading pathogen was Staphylococcus aureus (20%) while 73% were individuals who injected drugs. The likelihood of allograft infection after 20 years was 14% in patients with endocarditis and 5.6% in those without. Additionally, the endocarditis cohort had 41 explanted infected allografts, compared to 26 in patients without endocarditis. Risk factors for allograft infection in patients with endocarditis include earlier implants, intravenous drug use and younger age. In 18% of cases where patients had endocarditis and allograft infections, the causative pathogen was the same as the original organism.

To summarize, the long-term survival without reinfection significantly improved with the use of allogenic tissue. It is significant that most patients received an allograft for a cause not attributable to IE. Witten and colleagues [43] reported low infection rates in both patients without and with endocarditis, advocating continued use of allografts, particularly for invasive aortic root endocarditis treatment in modern times. While these outcomes are commendable, they should be interpreted with caution.

### 3.4. Alternative Valve Substitutes

The use of the Ross procedure is restricted for patients with infective endocarditis. This is due to the increased surgical complexity and potential long-term failure of two valves (aortic and pulmonary) when implanting the pulmonary autograft for aortic valve replacement. The outcome was disappointing, as the Ross procedure showed a three times higher operative mortality rate compared to conventional aortic valve replacement. As a result, the use of the Ross procedure declined significantly. However, studies have reported a relationship between volume and outcome, showing lower mortality in high-volume referral centers for IE treatment (0.3–1.1%) [45]. The Ross procedure has been re-evaluated in various publications, leading to a renewed interest in pulmonary autograft as an option for managing aortic valve endocarditis. Its use is recommended for specific patient populations in which avoiding the implantation of prosthetic materials is advised due to the heightened risk of recurrent infections or in women of childbearing age [84,85,86,87]. Substantial evidence has indicated that patients who receive a pulmonary autograft experience excellent long-term outcomes with lower pulmonic valve complication rates when faced with recurrent endocarditis [84,86,87]. Patients who have a life expectancy of over 15 years, a healthy and active lifestyle, and no significant comorbidities should be referred to surgical centers that possess significant experience [85]. Figure 1 (bottom panel) provides an image of a reconstructed CT of a Ross procedure conducted 23 years ago for endocarditis [87].

The most extensive report on aortic valve endocarditis shows a prevalent use of conventional mechanical valves. In cases of endocarditis complicated with abscess, in both normal and drug-abuser populations, Kim and colleagues [12] preferred mechanical prosthesis over stented xenografts; 40.5% versus 29.5% respectively. This trend was confirmed when both the intervalvular fibrosa and mitral valve were involved, and mechanical valves were used for surgical correction in 38% of patients, while 18.7% received a xenograft [11]. David et al. [11] achieved outstanding outcomes when utilizing mechanical valves alongside synthetic patches or prosthetic valve conduits in the treatment of complicated endocarditis of the aortic valve, which mandated the reconstruction of aorto-mitral discontinuity or aortic root. Nguyen et al. [49] found that the use of allograft resulted in poorer survival outcomes compared to stented xenograft bioprosthesis or mechanical valves (16%, 19%, and 65%, respectively) at a 5-year follow-up. The use of mechanical prosthesis was likely due to the younger age and reduced comorbidities in this group of patients.

Moon et al. [44] investigated the effect of selecting an appropriate valve substitute in the surgical management of left-sided endocarditis in patients who underwent surgery between 1964 and 1995. A total of 306 patients underwent surgery for left infective endocarditis (IE) with 68% of non-IV drug users and 32% of IV drug users received valve replacement. Among them, 62% received aortic valve replacement, 29% mitral valve replacement alone, and 9% had combined aortic and mitral valve replacement. During the initial period (1968–1976), mechanical valves were the preferred choice (61%) while stented xenograft bioprostheses were nearly exclusively used (98%) in the subsequent decade. In the intermediate period (from 1987 to 1995), surgeons’ inclination towards the use of bioprostheses (65%) exceeded the implantation of mechanical valves (25%) and allografts (10%). There were no disparities in operative mortality (18%) across different types of implanted valve. However, patients with native valve endocarditis (NVE) had notably better long-term survival than those with prosthetic valve endocarditis (PVE). Recipients of mechanical valves (2.1%) and bioprostheses (2.3%) faced the same risk of reoperation due to recurrence of infection after 5 years. Despite this, the risk of complications was lower for mechanical prostheses (0.5%) compared to stented xenografts (1.1%) over a 5-year period. Additionally, mechanical prostheses had better long-term survival rates, excluding operative death, after 10 and 20 years (62% and 46% respectively), in contrast to bioprostheses which had rates of 61% and 41%. Meta-analyses and observational studies comparing aortic allografts and conventional prostheses are provided in Appendix A [88,89,90,91,92,93,94,95,96].

## 4. Discussion

### 4.1. Systematic Approach to Treat Aortic Valve Endocarditis

Surgery aims to restore aortic valve function, while minimizing the risk of bacterial embolism during removal of the infected tissue [97,98,99,100,101,102,103,104]. Conservative surgery may be an option for patients with infection affecting only one leaflet (Figure 2 and Figure 3). To repair the aortic valve, it is necessary to remove any vegetation that may have accumulated, restore the correct alignment of the leaflets, repair any tears that may be present, and maintain the integrity of the aortic annulus [67,104].

The removal of infectious vegetation, also known as vegetectomy, is carried out along the cleavage plane of the leaflet (Figure 3, Type A1). To avoid tension on the suture line, it is advisable to reinforce the leaflet using a pericardial patch instead of direct suture of the lesion (Figure 3, Type A2). The Ozaki procedure [105] is a suitable option when the feasibility of valve repair depends on the limited extent of tissue damage (Figure 3, Type A3). Therefore, patients with a narrowed active infection but without valve devastation are more suitable candidates for treatment. Extensive damage to the aortic valve, particularly in the commissures and the development of an annular abscess, are major obstacles to aortic repair, and thus aortic valve replacement (Figure 3, Type B1 and 2) is the preferred course of action [46,47,92,93].

In patients with bicuspid aortic valve and annular dilatation, reconstruction is guided by specific criteria based on the presence of a single vegetation affecting 1 leaflet and 1 commissure. Excision of the infected portion can be repaired using a pericardial patch with 5-0 polypropylene suture. The annular stabilization is ensured with an annuloplasty ring, as per Lansac et al.’s description (Figure 3, Type A1–3) [106]. In cases where infection spreads to the mitro-aortic intervalvular fibrosa, which encompasses the A1 mitral scallop, anterolateral commissure, chordal apparatus, and left ventricular myocardium, it may be feasible to opt for a tricuspid allograft valve transplant (as shown in Figure 3, Type B3) [107,108,109,110]. In these cases, the initial site of infection typically affects the mitral valve leaflets, sometimes leading to an expanding lesion in the trigonal region and often accompanied by a large abscess in the mitral annulus. As the entire mitral valve and subvalvular apparatus become involved, valve function is severely compromised and may result in severe regurgitation.

### 4.2. Extended Infection: Replacement Surgery

Zao and colleagues [111] conducted a systematic literature review assessing morbidity and mortality rates in seven studies of infective endocarditis where valve repair or replacement procedures were employed. The researchers discovered significant differences in long-term survival outcomes in patients who underwent valve repair or replacement in three of these studies. For instance, one study compared aortic valve repair with replacement in IE. Improved: Patients who underwent aortic valve repair exhibited significantly better outcomes in terms of freedom from reoperation at 5 years (*p* = 0.021). In addition, the repair group showed improved 4-year survival rates compared to the replacement group (88% vs. 65%; *p* = 0.047). The second report indicated that patients who underwent MV repair had better event-free survival than those who received MV replacement after 10 years (*p* = 0.015). It is noteworthy that the MV repair cohort had a higher incidence of previous septic embolization. Furthermore, patients referred for mitral valve repair had superior outcomes as compared to those who received replacement (*p* < 0.05). An autonomous hazard factor for early and late demise (*p* < 0.05) was mitral valve substitution (Figure 3, Type B).

Mitro-aortic involvement is prevalent in infective endocarditis (IE). Feringa et al. [17] conducted a systematic review comparing the morbidity and mortality rates of mitral valve repair and replacement in IE patients. Results showed that patients who received valve repair had significantly lower in-hospital mortality (2.3% vs. 14.4%) and better 10-year survival rates compared to those who received valve replacement (7.8% vs. 40.5%). The systematic review [111] suggests a bias towards valve repair being superior to replacement in terms of endocarditis recurrence. However, valve repair is more commonly performed in cases of infective endocarditis that is localized to the mitral valve only. Although guidelines recommend mitral valve repair over replacement, when possible, in this category of patients [7,20], the feasibility of performing mitral valve repair is dictated by precise preoperative requirements and intraoperative findings [104,112].

Patients who have an extended AV infection that affects more than one leaflet and involves annular abscess or require PVE aortic replacement, may be considered for surgical intervention (see Figure 3, Type B). Aortic valve replacement can be achieved by using conventional stented xenograft, mechanical prostheses or implanting autologous or allogeneic tissue. However, there are various disadvantages associated with aortic valve replacement. The use of mechanical valves necessitates lifelong anticoagulation, which presents a significant disadvantage due to the risk of thromboembolism. Conversely, the main impediment to the use of stented xenografts and cryopreserved aortic allografts is the SVD, which can hinder their implantation. Greason and colleagues [113] found no significant differences in overall mortality and recurrence of infection over a 35-year follow-up between mechanical valve recipients and bioprosthesis recipients.

In cases of active infective endocarditis with substantial damage, including the mitral valve, double valve replacement is typically the preferred surgical option. However, to ensure the anchorage of the left ventricular wall to the valve apparatus, the chordae tendineae should be retained during replacement. The subvalvular mitral apparatus plays an essential role in tethering the chordae to maintain optimal mitral valve function. Removal of this apparatus leads to the loss of physiological tethering action by the chordae, resulting in severe long-term outcomes, especially in younger patients. Furthermore, early onset of heart failure can occur due to an increase in left ventricular wall stress accompanied by declining left ventricular function. MV sparing surgery is advised to avoid the disadvantage of chordal excision, which can lead to left ventricular enlargement and impairment (Figure 3, Type B1) [114].

STS Database analysis revealed that stented xenograft prosthetic valves are the preferred choice for active IE, with 8421 patients (73%) undergoing primary operations and 3139 patients (27%) undergoing reoperations. While allografts are infrequently used during primary operations, they are more commonly used for reoperations (Figure 2, top panel). Observational data over the past two decades indicate a significant decrease in allograft usage for first-time aortic valve replacement (from 9.4% to 5.6%) and re-operations (from 37.5% to 28.5%) according to the STS database (2005–2011). Allografts are more commonly used in re-operations than in primary interventions (32.2% vs. 7.0%, *p* < 0.0001) for valve (14.6%) and root re-placements (53.2%) [115]. Although Kim and colleagues [12] have raised questions regarding the use of allografts in non-extensive active endocarditis, this approach has been endorsed by numerous authors [104,109,116,117,118,119,120,121] and recently confirmed by Witten and colleagues [43]. These investigators have enhanced our understanding of how infection impacts the selection of allogeneic tissue in AVR and have brought attention to several critical clinical implications in complicated infective endocarditis.

### 4.3. Complicated Aorto-Mitral Endocarditis: Demolitive Surgery

Patients with positive Staphylococcus culture results are at an elevated risk of developing complicated and aggressive lesions, including aortic root abscess formation and involvement of intervalvular fibrosa and heart trigones. Moreover, such patients exhibit a significantly higher mortality rate compared to those infected by other pathogens. *E. faecalis*, a nosocomial opportunistic bacterium, is among the species that exhibit exceptional aggressiveness. The prevalence of this causative pathogen is high in cases of elaborate and complicated cardiac infections, emphasizing the necessity of alternative therapeutic methods such as immunotherapy or immunoprophylaxis. Antibiotics should be initiated promptly upon obtaining blood cultures. However, clinicians may opt for culture-guided therapy if the patient is stable [52,53,54,100,122]. Although empirical antibiotic regimens for treating native and prosthetic valvular endocarditis are based on guidelines established by the British Society for Antimicrobial Chemotherapy [100], resistance demonstrated by *E. faecalis* towards these treatments may necessitate surgical intervention, which may influence the selection of demolitive surgery. Intervalvular fibrosa extension, circumferential annular abscess, and the development of intracardiac fistulae are anatomopathological criteria that dictate the need for extensive removal and thorough cleaning of infected tissue. This process is followed by reconstruction using allogeneic tissues for both the aortic and mitral valves, as shown in Figure 3, Type C. It is important to adhere strictly to metric and unit standards throughout the process.

In our group, we have utilized total or partial mitral allografts with promising outcomes. At 5, 10, and 15 years, freedom from SVD was 90%, 76%, and 65%, respectively. Furthermore, freedom from reoperation reached the rate of 88%, 80%, and 64% at 5, 10, and 15 years, respectively. It is worth noting that patients with total allografts were more susceptible to the occurrence of SVD as mixed stenosis and insufficiency. SVD was more common in patients who had a non-endocarditis etiology and in those who had experienced pregnancy [123]. Similar evidence was presented in a series by Yankah and colleagues [120], where they implanted a mitral allograft, and in a study by Mestres et al. [108], where the use of mitral homograft in the tricuspid position was reported. The equivalence of allografts with bioprostheses in a group of young patients [117], as well as the convenience of these biosubstitutes for young women who plan to become pregnant in the future [124,125], provide strong support for allografts in this setting. Two technical aspects need to be emphasized. The aortic allografts were all put in place utilizing either the full root insertion or miniroot process. In relation to implant technique for mitral allografts, satisfactory outcomes were achieved through intraventricular fixation on the recipient papillary muscle, with a side-to-side positioning. This remained effective even in cases of largely infected and fragile mitral tissues [116,118].

We believe that this approach could decrease the risk of papillary muscle rupture, which other series have reported. The use of a monobloc aortomitral allograft, as described by Obadia and colleagues [126], may be a suitable choice for patients with infectious lesions in the intervalvular fibrosa and aortic root, depending on the severity and extent of the lesions [119,120]. Our group favors a double allograft replacement inserted as a separate block using either partial or total mitral allograft, as we believe this option minimizes the risk of size discrepancy compared to monobloc insertion and eliminates the need for left atrial roof reconstruction [79]. Our 8-year follow-up study indicates that adding a prosthetic ring to the mitral allograft is advisable for improved mitral replacement duration [117].

## 5. Risk of Allograft Infection

Several studies have indicated that infection is rare after allograft implantation, with lower risk observed among non-endocarditis patients compared to those with endocarditis during implantation. Additionally, Witten and colleagues [43] proposed that the risk of allograft infection was significantly associated with older donor age and younger patient age in the non-endocarditis group. Patients who had an allograft for endocarditis and underwent surgery in the early years of allograft use had a higher risk of premature reinfection. Furthermore, it was noted that late reinfections were linked to younger patient age and injection drug use. In the majority of cases, allograft reinfections occurred de novo, despite only 18% of infections being caused by the same pathogen. The predominant pathogens were Gram-positive cocci, such as *Streptococcus viridans* (22%), *Staphylococcus aureus* (20%), *Enterococcus faecalis* (10%), and Group D streptococci (11%), which are currently classified as enterococcal species. *S. aureus* notably caused infective endocarditis (IE) in 73% of injecting drug users, with 63% of cases caused by Gram-positive cocci linked to biofilm pili and exhibiting higher antibiotic resistance [51,52,53,54,55,56,57,58,59,60,61,62,63,64,65,66].

Despite a substantial decrease in the use of allografts for conditions other than endocarditis, various referral centers with substantial expertise in their utilization have compiled extensive case reports from 1990–2000. These centers prefer using allografts as an alternative for patients with smaller aortic roots as opposed to conducting “bio-Bentall” using conduits with a bioprosthetic valve. Skilled surgeons have developed a rescue approach to aortic root re-constructions by utilizing alternative valve replacements. This method offers benefits due to its pathophysiological reasoning, which provides excellent hemodynamic performance and eliminates the need for anticoagulation of the allogeneic tissue. Currently, the favored surgical approach utilizes modern bioprostheses, which enhance hemodynamics and act as superior platforms for transcatheter valve-in-valve methodologies. Less common alternatives comprise valve-sparing root replacement, Ross procedures, and Ozaki procedures [62].

In our study, we have discovered that the risk of infection arising immediately after allograft implantation for causes other than endocarditis is less than 0.2% per annum within the first decade. Furthermore, there is a 98.1% ± 1.5% freedom from endocarditis after two decades. Our data are in agreement with earlier research that shows a noteworthy absence of early peak risk of reinfection in patients who have received an allograft prosthetic valve. The patients exhibited a peak incidence of 1% to 1.2% per patient-year within 5 weeks. However, our study confirms a subsequent decrease in risk to 0.3% to 0.6% after 20 years. Our findings align with recent research published by Witten et al. The research carried out at the Cleveland Clinic found a notably low overall occurrence of allograft infection in patients with non-endocarditis etiology. This aligns with the overall endocarditis incidence observed in patients who receive aortic valve replacement with alternative prosthetic devices for non-endocarditis-related conditions.

Evidence from a registry based in Denmark (*n* = 16,018) suggested that patients who underwent implantation of a prosthetic aortic valve for causes other than endocarditis had a combined prevalence of endocarditis at 5 and 10 years of 3.4% and 5.2%, correspondingly. The Danish group also compared the incidence of endocarditis in subjects treated with transcatheter procedure (*n* = 632) versus those who had undergone standard surgical aortic valve replacement (*n* = 3777). They discovered that both groups had an equivalent 5-year cumulative occurrence rate (5.8% vs. 5.1%) [127]. A Society of Thoracic Surgeons database analysis, evaluating 39,199 patients above 65 years of age, found that mechanical valve recipients’ hospitalization rate for endocarditis over a 10-year span was 1.3%. Bioprosthetic valve recipients had a rate of 2%, with almost 1% experiencing endocarditis at implantation [115]. We postulated that late infections in patients with non-endocarditis causes are linked to allograft degeneration through two main risk factors: younger patient age and older allograft donor age. These factors hasten allograft degeneration, which can lead to structural dysfunction, creating an ideal environment for infection [39,128]. Similar conclusions were reached by researchers in Cleveland [31].

Landmark studies by Haydock and colleagues [37], alongside McGiffin et al. [38], which were published nearly two decades ago, suggested a lower risk of allograft infection in patients diagnosed with endocarditis. Further investigations revealed a lack of an early peak in recurrent endocarditis amongst patients who were treated with allograft as compared to those with conventional mechanical or stented xenograft valve prostheses. Hence, the conclusion was that allograft represents the ideal treatment for active endocarditis. These findings have been challenged by Witten and colleagues [43], who noted a protracted initial spike in re-infection rates followed by a decline in endocarditis patients. Although the literature demonstrates discrepancies in the recorded frequencies of early and delayed re-infection, Moon and his team [44] identified a heightened risk within the initial 5 years following graft, mechanical valve, and bioprosthesis implantation, regardless of the prosthesis type chosen. Sabik and colleagues [76] found that there is a peak occurrence of endocarditis between six and 18 months after allograft implantation while evaluating the risk of recurrent prosthetic valve endocarditis. Nevertheless, it is crucial to note that in most noticed contaminations, this peak resulted from fungal contamination of the allograft, as other studies have shown [39,129,130].

A noteworthy revelation from various studies is that there is a heightened risk of allogeneic tissue infection during the early phase, which appears to be influenced by the date of the surgical procedure. The risk is observed to be higher in cases of earlier dated allograft implants during a long-term follow-up of over twenty years. While certain factors, such as perioperative antibiotic usage, extent of debridement, buttress material, and suture technique, are not eligible for statistical analysis, it is conceivable that a better comprehension of endocarditis pathology gained over time, greater comfort with superior surgical radical debridement, and reduction in the use of prostheses and foreign materials during endocarditis surgery have significantly contributed to this enhancement. Two risk factors—younger recipient age of allografts and history of injection of drugs—appear crucial in the late recurrence of infection in allogeneic tissue. However, these factors are independent of each other. Thus, it remains a challenge to care for individuals who have used intravenous drugs and reduce opioid relapses, which is also related to managing the financial resources of the community in the post-patient discharge phase.

Several studies have reported that the use of allografts for endocarditis produces favorable results compared to conventional prosthetic valves. In the 1990s, patients with infective endocarditis who received conventional prosthetic valves experienced recurrences of up to 26% with invasive disease [131]. Jassar et al. [40] found a reinfection rate of 26% for mechanical composite grafts and 11% for biologic composite grafts at a 5-year follow-up. Similarly, David et al. [14] found that prosthetic valve recipients with paravalvular abscesses had recurrence rates of infection of 15% and 17% after 10 and 15 years, respectively. Kim et al. [12] found that the 5-year recurrence of reinfection was 17% for patients who underwent valve replacement with mechanical prostheses and 19% for those with biological prostheses. Sabik et al. [76] (*n* = 103) and Solari et al. [132] (*n* = 120) reported a 10-year recurrence of 3.7% and 5.0%, respectively, among proponents of allograft use. However, most studies investigating allograft reinfection have revealed slightly higher recurrences of 7.5% to 12% after 10 years [41,42,69]. The study by Wittman and colleagues [43] found that the cumulative incidence of allograft reinfection is lower than that noted in endocarditis prosthetic valve recipients in a large registry study from California and New York, recording a 12-year cumulative incidence of 9.4% for bioprosthetic recipients and 10% for those receiving mechanical prostheses [48].

Presently, we reserve the use of allografts when curative measures are needed for invasive endocarditis, preferring conventional prostheses for patients with a non-invasive infectious process. As per the 2017 guidelines from the American Association for Thoracic Surgery for surgical treatment of endocarditis, allograft root replacement is recommended for those diagnosed with aortic valve invasive endocarditis. Observations suggest that the use of allogeneic tissue is preferable, particularly in cases where the infection is severe and extensive, over alternative ducts with valve prostheses as a more robust option [19,20].

The benefit of utilizing a selective and differentiated allograft method, instead of the conventional approach that involves prosthetic valve implantation with or without a prosthetic conduit, has not been strongly validated in our research [13,39,67,79] nor in other reports [43,75,76,133,134]. However, both our research [13,39,67,79] and that reported by other groups [43,75,76,132,133] demonstrates the advantages of a strategy that guarantees radical debridement and the implementation of allograft root replacement in the treatment of invasive endocarditis. This method can yield outstanding outcomes with a minimal chance of allograft reinfection. Similarly, several studies have demonstrated a high mortality rate following allograft implantation in patients with invasive endocarditis. This limitation reduces the effectiveness of using allograft with a full root replacement technique [70,71,78].

## 6. Limitation

The main issue with the IE studies examined in this review is that, in most cases, after aortic valve replacement with traditional allografts or prostheses, infection or reinfection occurred over time, as estimated by event probability methods such as the Kaplan–Meier estimator. It is worth mentioning that study cohorts can be more intricate, and the populations evaluated are not always uniform. Thus, in statistical analysis, the Kaplan–Meier estimator typically ignores the potential censoring of information from competing risks of death and valve dysfunction explanations unrelated to endocarditis. This can result in overestimating the risk to patients whose deaths stem from causes other than device-related infection and underestimating the inherent susceptibility of these devices to infection. Only one of the recently reported studies [43] provides results according to the conventional probability of the event, with two different estimates derived from the analysis of competing risks, specifically cumulative incidence and conditional probability. Furthermore, this study focused on the risk of infection that is inherent in the allograft. Wittmann and colleagues’ research indicates that endocarditis implants have a higher risk of allograft infection compared to implants for other indications. Over time, this difference has only increased, and it is evident that the increased risk of reinfection does not disappear, but is more prevalent as the allografts become increasingly susceptible to infection. Importantly, the study’s limited scope, with only one surgeon, restricts the generalizability of the findings for those outside the field of aortic root reconstructive surgery utilizing aortic allograft. Finally, the lack of comprehensive outcomes, obtainable through a follow-up period representing no less than 95% of the total follow-up time, can be established by the absence of longitudinal echocardiographic data. In the absence of these data, any association between allograft usage, inherent risk of allograft infection after AVR, and valve performance cannot be determined.

## 7. Conclusions

Surgery aimed at treating infective endocarditis (IE) of the aortic valve involves assessing lesion characteristics in a comprehensive and systematic manner. There are several clinical and pathological variables that can affect postoperative survival rates. Surgery is indicated and timed in such a way as to quickly restore aortic valve function, minimize the risk of embolization, and prevent deterioration that could lead to heart failure. Various factors determine the type of surgical approach required, such as the underlying cause (i.e., noninvasive or invasive organism), the size of the infectious lesion, whether a native or prosthetic valve is involved, the age of the patient, and any additional medical conditions, including any cardiac or extracardiac organ dysfunction. Evidence of complicated endocarditis extending to the intervalvular fibrosa, involving the mitral annulus, and presenting with a fistula requires extensive surgical intervention, utilizing allografts and/or synthetic materials. Patients who have undergone an allograft implant for endocarditis experience low occurrences of infection both in the short and long term. In patients with endocarditis, earlier implantation surgery, younger age, and recreational drug use elevate the risk of infection. This review presents estimates of the inherent infection risk in patients with endocarditis who received an allograft implant due to root invasive endocarditis. This analysis initiates further discussion regarding the usefulness of this allogeneic conduit. Moreover, significant evidence arises from this study concerning the infection risk in allograft utilization in patients without an endocarditis cause. The use of allografts has been proven to be safe and effective in patients without endocarditis pathologies. However, in this particular group of patients, older allografts and younger patients were identified as risk factors for endocarditis.

## Figures and Tables

**Figure 1 life-13-01980-f001:**
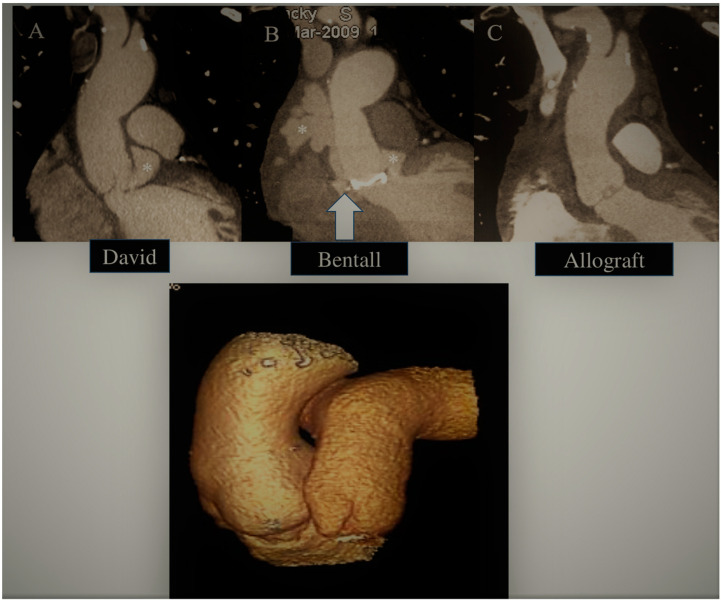
**Top**: Recurrent false aneurysm (asterisk and white arrow) due to endocarditis following (**A**) David and (**B**) Bentall procedures treated with (**C**) aortic allograft replacement. **Bottom**: Ross operation at 23 years follow up.

**Figure 2 life-13-01980-f002:**
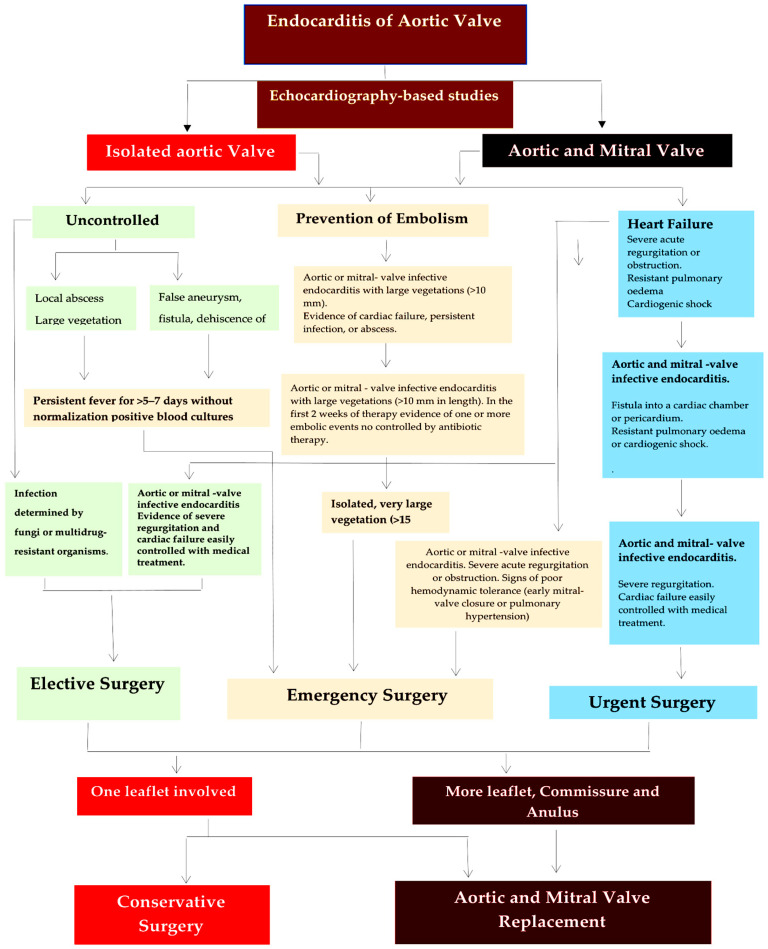
Decision tree algorithm for the management of aortic valve endocarditis and extended IE involving the intervalvular fibrosa and mitral valve. Treatment for IE depends on the responsible bacterium. Surgery is recommended as elective (green box), emergency (rose box), or urgent (blue box). Repair or replacement of the aortic valve alone or both the aortic and mitral valves can be performed in extended IE cases. Conservative surgery with aortic valve repair can be considered for a localized infection of the valve leaflet (light red box). For severe infection and significant anatomical damage, it is recommended to surgically replace the aortic valve (dark red box) with timing determined by a shared decision amongst multiple disciplines. In the emergency surgical option (rose box), the infected valve should be removed within 24 h of diagnosis completion. In the urgent surgical option (blue box), surgery should be carried out within a few days of the indication being given. For patients opting for elective surgery (green box), the procedure should not be performed until 1 to 2 weeks after receiving antibiotics.

**Figure 3 life-13-01980-f003:**
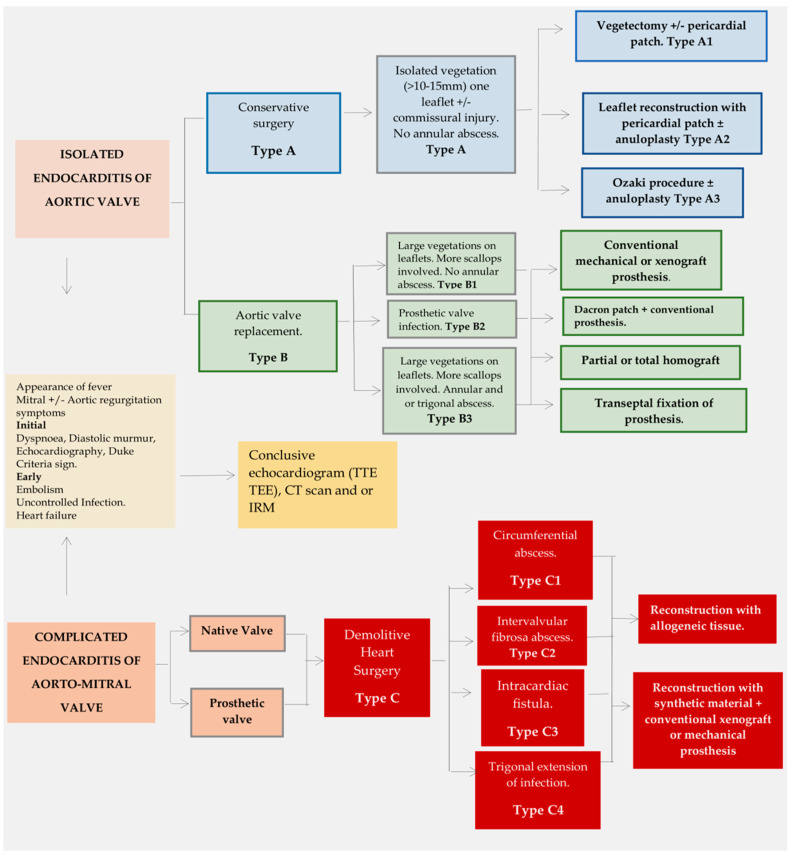
Decision algorithm for diagnosing and treating aortic valve endocarditis. TEE ultrasound provides improved diagnostic sensitivity and specificity for identifying the presence, location, and extent of vegetations. Further diagnostic measures such as cerebral MRI, CT with or without PET–CT, [FDG] PET-CT, or leucocyte-labeled single SPECT CT scan, MRI, PET, and SPECT are suitable for detecting complicated lesions in active endocarditis. The document highlights three standardized approaches, each outlined in a separate box. For type A (blue box), the conservative option (type A1) is recommended if there is a localized lesion without annular abscess. The repair consists of vegetectomy (type A1), reconstruction with an autologous or synthetic pericardial patch (type A2), or Ozaki procedure (type A3). In Type B (green box), indicators for tailored aortic valve replacement include compromise of the entire valve (Type B1), involvement of the annulus (Type B2), or the presence of an abscess extending to the trigones (Type B3). Mechanical and biological prostheses may be implanted with or without the addition of a prosthetic conduit or a Dacron patch. In Type C (red box), the shared decision-making process considers demolitive surgery for native valve endocarditis (NVE) and prosthetic valve endocarditis (PVE). Reconstruction using allograft or conventional prosthetic materials (patches and/or prosthetic valve conduits) is necessary for managing circumferential annular abscess (type C1), mitral-aortic intervalvular fibrosa involvement (type C2), atrial fistula (type C3), and impairment of the trigones (type C4). (From Nappi and colleagues [13,67] Abbreviations: AV (aortic valve), CT (computed tomography), FDG (fluorodeoxyglucose), MRI (magnetic resonance imaging), PET (positron emission tomography), SPECT (photon emission CT), TEE (transesophageal echocardiography).

**Table 1 life-13-01980-t001:** Narrative review searching strategies.

Items	Specification
Date of Search (specified to date, month, and year)	From February 2023 to June 2023
Databases and other sources searched	MEDLINE, Embase, and the Cochrane Library
Search terms used (including MeSH and free text search terms and filters)	“endocarditis” or “infective endocarditis” together with “allograft”, “homograft”, “allograft endocarditis”, “pathogenesis”, “manifestations”, “treatment”, and “surgery”
Timeframe	Up to June 2023
Inclusion and exclusion criteria (study type, language restrictions etc.)	English language
Selection process	Two authors independently selected articles after screening for duplicates

## Data Availability

Not applicable.

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
