# Peer review of "A Literature Review on the Use of Aortic Allografts in Modern Cardiac Surgery for the Treatment of Infective Endocarditis: Is There Clear Evidence or Is It Merely a Perception?"

_life, 2023, doi:10.3390/life13101980_

Round 1
Reviewer 1 Report
Endocarditis is a complex pathology, in which the complications that arise often require surgical treatment
Due to the limited experience with the use of the allograft, there is no clear evidence of its superiority, although in the case of destruction of the aortic ring, and periannular abscesses, these have proven effective in many cases.
This review is a complex study, which could have a significant contribution to the approach and surgical techniques in endocarditis
Author Response
The authors express their gratitude to the reviewer for the feedback provided.
Attached is the report for other reviewers.

Author Response
The authors express their gratitude to the reviewer for their feedback. Please find the responses to the comments enclosed in the attached PDF document.

Reviewer 3 Report
The author has reviewed a substantial amount of literature and has compiled clinical outcomes and current recommendations for the use of allograft in surgery for aortic valve infective endocarditis. The article is informative and interesting, but I have a few suggestions that may make it easier for readers to understand and engage with the content.
1. The article is quite lengthy. While Life does not impose strict word limits, a review article exceeding ten thousand words, even after excluding references, can be overwhelming for readers. This length may distract readers from the key point the author aims to convey, which is the use of allograft in surgery for infective endocarditis patients. Additionally, excessively long articles can lead to errors in the author's narrative. For instance, in lines 188-191, "In support of this hypothesis we identified mainly gram-positive pathogens suggesting the crucial role offered by proteins of the family of adhesive matrix molecules (MSCRAMM) that may serve as potential antigenic candidates for the development of new promising immune-therapies," this sentence appears to be directly copied from reference 61. However, in this context, it should be rewritten to better fit within the article.
2. The author has created elaborate algorithms to help readers summarize conclusions. However, these algorithms are overly complex and tend to reflect the author's personal opinions, which can confuse readers. For instance, in Figure 2, the column stating "aortic or mitral valve IE. Evidence of severe regurgitation. Cardiac failure easily controlled with medical treatment" can apply to both elective surgery and urgent surgery, making it difficult to understand the algorithm. Furthermore, in Figure 2, the column for "mitral valve replacement" should perhaps be "valve replacement" since the figure legend mentions "surgical replacement of the aortic valve." I also recommend avoiding definitions for elective/urgent/emergency surgery that differ from common understanding, as this can lead to confusion when reading the algorithms.
3. Since this is a review article, I suggest including the Supplementary table in the manuscript. This would make it easier for readers to understand the author's points, and the table is not too large to be accommodated within the manuscript.
Overall, these suggestions aim to enhance the readability and clarity of the article, making it more accessible and engaging for readers.
Author Response
The reviewers are thanked for their feedback. Extensive revisions have been made, including the addition of tables to the main text for the final version.
The attached report has been sent to other reviewers.

Round 2
Reviewer 2 Report
Line 372 correct Figure 1